# Clinician-recalled quoted speech in electronic health records and risk of suicide attempt: a case–crossover study

Lasantha Jayasinghe ,[1] André Bittar,[1] Rina Dutta,[1,2] Robert Stewart[1,2]

RD and RS contributed equally.

[1]Institute of Psychiatry, Psychology and Neuroscience, King's College London, London, UK
[2]South London and Maudsley NHS Foundation Trust, London, UK

**Correspondence to**
Lasantha Jayasinghe;
lasantha.jayasinghe@kcl.ac.uk

## ABSTRACT

**Objective** Clinician narrative style in electronic health records (EHR) has rarely been investigated. Clinicians sometimes record brief quotations from patients, possibly more frequently when higher risk is perceived. We investigated whether the frequency of quoted phrases in an EHR was higher in time periods closer to a suicide attempt.

**Design** A case–crossover study was conducted in a large mental health records database. A natural language processing tool was developed using regular expression matching to identify text occurring within quotation marks in the EHR.

**Setting** Electronic records from a large mental healthcare provider serving a geographic catchment of 1.3 million residents in South London were linked with hospitalisation data.

**Participants** 1503 individuals were identified as having a hospitalised suicide attempt from 1 April 2006 to 31 March 2017 with at least one document in both the case period (1–30 days prior to admission) and the control period (61–90 days prior to admission).

**Outcome measures** The number of quoted phrases in the control as compared with the case period.

**Results** Both attended (OR 1.05, 95% CI 1.02 to 1.08) and non-attended (OR 1.15, 95% CI 1.04 to 1.26) clinical appointments were independently higher in the case compared with control period, while there was no difference in mental healthcare hospitalisation (OR 0.99, 95% CI 0.98 to 1.01). In addition, there was no difference in the levels of quoted text between the comparison time periods (OR 1.09, 95% CI 0.91 to 1.30).

**Conclusions** This study successfully developed an algorithm to identify quoted speech in text fields from routine mental healthcare records. Contrary to the hypothesis, no association between this exposure and proximity to a suicide attempt was found; however, further evaluation is warranted on the way in which clinician-perceived risk might be feasibly characterised from clinical text.

## INTRODUCTION

Around 800 000 deaths a year internationally are estimated to be the result of suicide.[1] In the UK, 5821 deaths were attributed to suicide in 2017.[2] Given that predictions of suicide risk have not improved significantly in the last 50 years, new data-driven methods would

### Strengths and limitations of this study

► A larger sample size (1503 patients) than previous related studies.
► The case–crossover design eliminates between-patient differences as potential confounding factors, making the results more robust.
► Time-varying factors that could potentially cause confounding were included as covariates in the regression.
► 90% of the 14 960 patients making a suicide attempt in the case period could not be analysed, due to no documentation in the control period, although analysis appears representative.

be welcomed, and a recent meta-analysis concluded a shift away from risk factors to risk algorithms.[3 4]

Electronic health record (EHR) data provide important potential opportunities in this respect, and for patients receiving mental health services they hold a rich source of longitudinal data leading up to recorded suicide attempts. There are also benefits over traditional assessment methods such as questionnaires, in which respondents may not answer accurately due to perceived stigma[5–7] or limited recall.[5] Commonly, predictive studies have sought to understand suicide risk in terms of the structured data within EHRs, such as diagnostic codes.[8 9] However, recent studies have begun to examine the value of including unstructured EHR data for predicting suicide risk. For example, text-mined EHR data were found to produce more accurate models of suicidal behaviour in a sample of Gulf War veterans,[10] and important information such as recorded suicidal ideation and past attempts has been successfully identified in mental health-care text using natural language processing (NLP).[11]

Previous studies using EHRs to investigate suicide risk have focused on identifying individuals at risk of suicide from those not at

risk (ie, between-person variation). These have included Bayesian modelling to identify patients at risk of suicide attempt where patients had made three or more healthcare visits in a retrospective cohort,[12] Cox regression to develop a 10-year probability prediction model for death by suicide in a sample from the Korean National Health Insurance Service[13] and neural networks applied to the EHRs of UK patients to identify those most at risk of suicide.[14] Simon and colleagues[15] examined the records of US patients across seven health systems and determined that incorporating EHR data provided significant improvements to existing suicide prediction methods, while Karmakar and others[16] developed a novel model using physical illness data from the EHR to predict suicide attempts. Walsh and others[9] used machine learning approaches to successfully identify patients who made a suicide attempt from those that self-harmed. While this between-person investigation is an immensely important area of research, relatively little investigation has taken place using EHRs to investigate temporal changes associated with suicide risk (ie, within-person associations), although it is known, for example, that risk is relatively high in relation to particular clinical events (eg, shortly after discharge from inpatient care[17]), as well as showing seasonal fluctuation.[18] The case–crossover design[19] was introduced as a method for studying transient effects on the risk of acute events. Previous research has employed this design to study suicidal behaviours in terms of negative life events,[20] death of close relatives,[21] substance abuse[22] and antidepressant drugs.[23 24] These studies gathered information from structured fields in registers, questionnaires and insurance records, but not directly from the free text in EHRs. Therefore, this study sought to determine transient changes within a patient's record over time that might contribute to the patient's risk of suicide.

Considering the source data for this study, the tendency has been to view the EHR as simply a factual summary of a patient's experience and symptoms. However, it is important to bear in mind that the EHR can also be viewed as a narrative account written from the perspective of the clinician and other healthcare professionals.[25] Clinician reporting in the EHR varies widely and the nature/style of the reporting may provide additional information beyond what is actually recorded.[26] A frequent comment in EHRs is a generic phrase such as 'No evidence of suicidal ideation', yet this raises the question of whether the clinician actually asked about suicide or just observed for signs indicating suicidality. A video analysis of outpatient visits in one study found that most questions about suicidal ideation were closed yes/no questions, and three-quarters of the questions were negatively phrased, inviting patients to confirm they were not feeling suicidal.[27]

Trainee psychiatrists are often taught to use brief verbatim patient quotes to record a patient's actual words in order to provide better evidence for their decision-making and also as part of medical defensive practice.[28 29] In EHRs this may manifest in direct quotations of statements made by the patient, or by 'referencing'—where a clinician assigns the source of the text to someone other than themselves. Associations of recording style with perceived risk are supported by a higher relative frequency of third-person pronoun use found in groups of veterans who had died from suicide compared with a group of service users who were still alive.[30] In that study, the use of a group of words related to referencing did not vary significantly between the two groups, but the authors did not report on directly quoted speech in quotation marks. However, another investigation found that quoting 'he/she says' increased in records of clinician–patient interactions that involved communication of bad news between doctor and patient.[31]

Clinical recording style has received little investigation in the field of suicidology. To our knowledge, research into the use of quotation marks to record verbatim patient statements in notes has not been conducted to date. We therefore sought to investigate the hypothesis that the frequency of quoted phrases in an EHR would be indicative of higher perceived immediate risk, in that they would be more frequent in time periods closer to a suicide attempt. Assuming that this reflects a transient change within a patient from one time period to another, a case–crossover design was used in this study.

## METHODS
### Study sample
The mental health records used in this study were deidentified copies of those from the South London and Maudsley (SLaM) NHS Foundation Trust, assembled using its Clinical Record Interactive Search (CRIS) platform, which currently accesses mental healthcare records for over 400 000 patients.[32] SLaM provides comprehensive, near-monopoly mental healthcare services to a geographic catchment of around 1.3 million residents in four boroughs of South London, as well as some national specialist services. CRIS data have been linked to the Hospital Episode Statistics (HES) database, which contains details of all admissions, accident and emergency attendances and outpatient appointments at NHS hospitals in England.[33] This CRIS–HES linkage was used to determine suicide-related hospital admissions for individuals between 1 April 2006 and 31 March 2017 inclusive, identifying 14 960 unique patients with at least one suicide attempt, indicated by the presence of any of the following International Classification of Diseases codes: X6*, X7*, X80-4*, Y1*, Y2*, Y30-4* and Y87*, associated with a hospitalisation lasting at least 24 hours (ie, starting and ending on different dates). From these, a subset of 1503 patients (10.0%) were identified who had at least one document from mental healthcare in both the case and control periods (see Case-crossover design section) and who thus provided sufficient data for the analysis. Of these 1503 patients, 877 (58.3%) have at least one quotation in the control period, 919 (61.1%) have at least one quotation in the case period and 625 (41.6%) have at

least one quotation in both the control and case periods. In addition to the text-derived data, demographics including age, gender and ethnicity were extracted to describe the cohort.

## Patient and public involvement

We did not directly incorporate patient and public involvement (PPI) into this particular study but the SLaM Biomedical Research Centre Case Register used in the analysis was developed with extensive PPI and is overseen by a committee that includes service-user representatives.

## Case–crossover design

In this study the case–crossover design was used to compare the occurrence of quoted phrases in clinical text between the period just prior to a suicide attempt and a control period, within the same individual. The advantage of this design is that although comparisons cannot be made between individuals, individual confounders that do not vary over time, such as gender, race and genetics, are eliminated.[34] Additionally, this design is particularly well suited to EHR data research which allows for sufficiently large samples experiencing a given event, which is not possible in the traditional cohort study; also EHR data do not depend on recollection of past events, which may lead to recall biases. On the other hand, the design requires data within appropriate comparison periods and may therefore need to be restricted to clinical subgroups where this is present; in addition, other time-varying factors may act as confounders and need to be captured and quantified. For the analyses presented here, the index date was the hospital admission date for a first suicide attempt. The case period was defined pragmatically and a priori as 1–30 days prior to the index date and the control period was set at 61–90 days prior to this date.

## Time-variant factors

Considering other potential differences between the two comparison time periods, the following variables were considered as confounding factors and included as covariates in the regression: (1) number of face-to-face outpatient appointments attended, (2) number of appointments made and not attended, (3) number of inpatient bed-days in SLaM. These were all included as potential confounders, as differences in these variables in the two time periods would result in differences in the number of documents and therefore the number of quotations present per patient that might have been caused by factors other than the period effect.

## Identification of quoted speech

An NLP tool was developed using regular expression matching to identify text occurring within quotation marks in the EHR. To avoid mistaking apostrophes used in contractions for the start of quoted phrase, a quote followed by a sequence ('c', 'd', 'e', 'm', 'n', 's', 't', 've', 're', 's', 'll', 'all') was treated as an apostrophe, not a quote. A similar check was performed for end quotes. Once a quoted phrase was identified, any subquotations occurring within that quote were assumed to be part of the larger quotation. The length of quoted phrases was allowed to vary from one word to more than a paragraph; however, a maximum length of 1500 characters was applied to avoid extracting the entire text where a quote was not properly closed. Phrases that consisted only of emails or URLs were removed using standard regular expression pattern matching and substitution procedures.

The performance of the algorithm was tested on random samples of both clinical correspondence and case note documents from CRIS (table 1), with a precision of

**Table 1**  Test set performance metrics for random documents selected to contain the quotation mark characters stated

| Quotation mark characters contained | Documents (n) | Quotations (n) | Precision | Recall | F score | Accuracy |
|---|---|---|---|---|---|---|
| Clinical correspondence | | | | | | |
| " | 10 | 111 | 0.96 | 0.95 | 0.95 | 0.91 |
| " | 10 | 50 | 0.98 | 0.98 | 0.98 | 0.96 |
| " | 9* | 35 | 1.00 | 1.00 | 1.00 | 1.00 |
| None of: ", ", " | 10 | 4 | 1.00 | 1.00 | 1.00 | 1.00 |
| Case notes | | | | | | |
| " | 10 | 59 | 0.98 | 0.92 | 0.95 | 0.90 |
| " | 10 | 53 | 0.98 | 0.92 | 0.95 | 0.91 |
| " | 10 | 18 | 1.00 | 1.00 | 1.00 | 1.00 |
| None of: ", ", " | 10 | 0 | – | – | – | – |
| At least one of: ', ', ", ", " | 10 | 10 | 0.91 | 1.00 | 0.95 | 0.91 |
| All of: ', ', ", ", " | 10 | 91 | 0.99 | 0.98 | 0.98 | 0.97 |
| All of: ', ', ", ", " and document length >1000 characters | 30 | 264 | 0.97 | 0.92 | 0.95 | 0.90 |
| Total | 129 | 695 | 0.98 | 0.95 | 0.96 | 0.92 |

Minimum document length set to 500 characters, unless otherwise stated.
*One document removed as it was textually corrupt.

0.98, recall of 0.95 and F score of 0.96 across the whole test sample. To test the performance in the model sample, 15 random documents identified by the algorithm as containing quotations and 10 without quotations were examined in both the case and control periods. In summary, 49 documents were evaluated, yielding a precision of 0.92, recall of 0.93, F score of 0.92 and accuracy 0.86. One document was eliminated from the analysis due to incorrect syntax. The majority of inconsistencies between the algorithm and annotated quotations were due to grammatical inconsistencies such as omitting one of the pair of quotation marks or mismatching two different quotation mark styles. This was not amended in the regular expression matching pattern as it was felt to be more likely to produce false positives than genuine quotes. It is important to note that the algorithm was not designed to identify the speaker of the quoted text; however, analysis of a random sample of 25 documents each from the control and case periods showed that quotations referred to patient speech in 70.5% and 95.8% of instances, respectively.

## Statistical analysis

Data were cleaned using standard Python (V.3.6.8) libraries. Statistical analyses (paired t-tests and conditional logistic regressions) were performed in R (V.3.6.0). Two-tailed paired t-tests were carried out to investigate differences in the number of quoted phrases between the control and case periods. Quotations per token were used to normalise for document length. Main analyses used conditional logistic regression to examine the association between the number of quoted phrases in the control and case periods.

## RESULTS
## Cohort characteristics

The demographics of the study cohort and the wider sample of patients with at least one suicide attempt in the time period of analysis are described in table 2. Of the 1503 individuals in the cohort the majority were female, the mean age at first presentation to SLaM was 34.8 years (SD=16.1 years), with 64.5% aged 40 or under. The majority were of White European background, and the next largest ethnic group was Black. These characteristics of the study cohort were broadly similar to those of the wider sample (table 2). Individuals in the study cohort had approximately the same number of documents in the control period (mean=14.7, SD=28.5) and case period (mean=15.3, SD=28.9); p=0.381.

## Univariate analyses

Univariate analyses of the individual covariates were carried out to investigate differences between the case and the control periods (table 3). The univariate distributions were non-normal, displaying high peaks and long tails; however, they were compared with paired t-tests, which are robust under this distribution.[35] Face-to-face

**Table 2** Demographic characteristics of the study sample (n=1503) in comparison to all patients with hospitalised suicide attempt (n=14 960)

| Demographic variables | Study sample | | All suicide admissions | |
|---|---|---|---|---|
| | n | % total | n | % total |
| Gender | | | | |
| Female | 932 | 62.0 | 8463 | 56.6 |
| Male | 571 | 38.0 | 6488 | 43.4 |
| Unknown | 0 | 0.0 | 9 | 0.1 |
| Ethnicity | | | | |
| White European | 1083 | 72.1 | 9805 | 65.5 |
| Black | 274 | 18.2 | 1512 | 10.1 |
| Asian | 66 | 4.4 | 615 | 4.1 |
| Other | 66 | 4.4 | 756 | 5.1 |
| Unknown | 14 | 0.9 | 2272 | 15.2 |
| Age at index hospitalisation (years) | | | | |
| <16 | 184 | 12.2 | 1612 | 10.8 |
| 16–20 | 205 | 13.6 | 2204 | 14.7 |
| 21–30 | 271 | 18.0 | 3567 | 23.8 |
| 31–40 | 310 | 20.6 | 2935 | 19.6 |
| 41–50 | 280 | 18.6 | 2615 | 17.5 |
| 51–60 | 156 | 10.4 | 1195 | 8.0 |
| 61+ | 97 | 6.5 | 829 | 5.5 |
| Unknown | 0 | 0.0 | 3 | 0.0 |

outpatient attendances and non-attendances were significantly higher in the case period compared with the control period, but there was no significant difference in the number of inpatient bed-days. With regard to the exposure of interest, no difference was found in the number of quoted phrases for the patient group in the case period compared with the control period.

## Conditional logistic regression

Univariate and multivariate conditional logistic regression results are presented in table 4. Considering presence or not of quoted text as a binary variable, although an association of borderline significance was present in the unadjusted model (OR 1.17, 95% CI 0.99 to 1.38), this was attenuated substantially after adjustment for other covariates (OR 1.09, 95% CI 0.91 to 1.30) with the majority of the attenuation occurring following adjustment for level of face-to-face contact. The full details of the coefficients in the quotations binary model, including face-to-face contacts, appointments not attended and inpatient bed-days as covariates, are presented in table 5. Associations between number of quoted text instances and case versus control periods were close to the null in all models, while both attended (OR 1.05, 95% CI 1.02 to 1.08) and non-attended (OR 1.15, 95% CI 1.04 to 1.26) appointments were independently higher in case compared with control periods.

**Table 3** Univariate analyses—paired t-test results

| Variable | Control period | | Case period | | Difference | 95% CI | P value |
|---|---|---|---|---|---|---|---|
| | Mean | SD | Mean | SD | | | |
| Face-to-face contact | 2.47 | 3.71 | 2.92 | 4.14 | 0.441 | 0.225 to 0.657 | <0.001 |
| Appointment not attended | 0.38 | 1.00 | 0.47 | 1.13 | 0.098 | 0.036 to 0.161 | 0.002 |
| Inpatient bed-days | 2.29 | 7.13 | 2.05 | 6.69 | −0.242 | −0.620 to 0.134 | 0.207 |
| Quotations per token* | 0.16 | 0.25 | 0.16 | 0.23 | −0.0004 | −0.0158 to 0.0149 | 0.956 |

*Tokens refer to word tokens as determined by the Python nltk Regex Word Tokenizer.

## DISCUSSION

To our knowledge, this was the first study using a case–crossover design to examine whether changes in the frequency of clinician-recalled quoted text might serve as an indicator of suicide risk in patients. In summary, we found no association between the frequency of quoted phrases between periods close to and less close to the occurrence of an attempted suicide event, although there were differences in other metrics between the time periods, notably increased numbers of both attended and non-attended appointments closer to the event in question.

Previous research has been sparse on mental health risk implications of different reporting styles in clinical records. As mentioned, one previous study had reported no association between referencing text as a construct and suicidality[30]; however, we felt that the association of text in quotation marks was worth evaluating as an alternative marker, because of the potential strengths of a case–crossover design, and because of another study's findings that quoted text used in particular clinical circumstances is associated with higher perceived risk.[31] One possible difference between that study and ours is that we used quotation marks to identify quoted phrases, rather than attempting to identify the speaker, or exchanges of views between clinician and patient. Thus, extending the definition of quoted text to the identification of relevant pronouns would be a worthwhile avenue for future research.

Additional findings of our study were that increased face-to-face contact and failure to attend appointments were significantly more common in the case compared with control period and thus were markers of higher risk status. These are clinically plausible associations and support the robustness of the case–crossover design for this outcome. Patients with increased face-to-face contact in the case period would presumably have a greater number of documents and potentially quotations than in the control period, while this would be less likely in those failing to attend appointments in the case period; supporting this, adjusting the association of interest for face-to-face contacts resulted in substantial attenuation, whereas adjusting for non-attendances made little difference to coefficients.

The data for this study were drawn from a much larger source than the previous studies: for example, a narrative analysis of four medical interviews[31] or a sample of 63 veterans with the equivalent number of controls.[30] Another key strength of this study is that between-patient confounding factors are eliminated through the case–crossover design, which renders a more robust analysis than that attempted previously, and the positive associations with attended and non-attended appointments

**Table 4** Conditional logistic regression models for the association between levels of quoted speech and time period prior to hospitalised suicide attempt

| Characteristic | OR | 95% CI | P value |
|---|---|---|---|
| Total sample (n=1503) | | | |
| Quotations_binary | | | |
| Unadjusted | 1.17 | 0.99 to 1.38 | 0.073 |
| Adjusted for face-to-face contacts | 1.08 | 0.90 to 1.28 | 0.411 |
| Adjusted for DNA | 1.16 | 0.98 to 1.37 | 0.087 |
| Adjusted for number of inpatient bed-days | 1.18 | 1.00 to 1.40 | 0.050 |
| Adjusted for all covariates | 1.09 | 0.91 to 1.30 | 0.346 |
| Quotations_per_token* | | | |
| Unadjusted | 0.99 | 0.71 to 1.38 | 0.956 |
| Adjusted for face-to-face contacts | 0.93 | 0.67 to 1.31 | 0.693 |
| Adjusted for DNA | 0.99 | 0.71 to 1.39 | 0.971 |
| Adjusted for number of inpatient bed-days | 0.99 | 0.71 to 1.39 | 0.969 |
| Adjusted for all covariates | 0.94 | 0.67 to 1.32 | 0.735 |

*Tokens refer to word tokens as determined by the Python nltk Regex Word Tokenizer.

**Table 5** Full conditional logistic regression model for covariates associated with time period prior to hospitalised suicide attempt

| Characteristic | OR | 95% CI | P value |
|---|---|---|---|
| Total sample (n=1503) | | | |
| Quotations (binary) | 1.09 | 0.91 to 1.30 | 0.346 |
| Face-to-face contacts | 1.05 | 1.02 to 1.08 | 0.001 |
| Appointments not attended | 1.15 | 1.04 to 1.26 | 0.004 |
| Inpatient bed-days | 0.99 | 0.98 to 1.01 | 0.211 |

support the robustness of the time period comparison. As well as difference in individual-level characteristics, the design should also have equalised most clinical/ service-related potential confounders, such as workflow, providers and system-level factors, particularly as the comparison time periods are relatively proximal to each other and are unlikely to contain substantial differences in service type. However, it is important to bear in mind that the case–crossover design only addresses within-individual variations as a risk factor, and clearly cannot be used to test associations with characteristics that vary between individuals.

A limitation of our study is that we were only able to analyse a small sample of the 14960 patients who had made a suicide attempt during the time period of interest, 79% of whom had no prior contact with SLaM (ie, no documents at all in the case and control periods), although those analysed appeared representative of the source sample on metrics investigated. Consequently, the findings are limited to patients who were already seeking mental health treatment. Previous research shows that in a geographically diverse study of people dying by suicide in the USA, in the month prior to death, only 5% of cases received psychiatric treatment, but over 60% had made primary and secondary care visits in the preceding year.[36] Potential linkage of information beyond mental health-care to primary and acute care might prove useful for predictive modelling, in the mining of text and through a wider variety of structured information on the nature and level of service contact.

The absent association of interest might possibly have arisen because the use of quotations is a relatively invariant factor based on individual clinician linguistic style; it was not possible to account for this limitation in the analysis and this is important as different individuals may have differing tendencies to quote patient speech. Another limitation was that we were not able to determine the distance between the recording of speech as it is produced and when the transfer electronically occurred. It is common practice for clinicians to handwrite notes and transfer them to electronic records later, so we cannot precisely determine if quotations were actually produced in the case or control periods, and this may have affected the outcomes. Additionally, this algorithm did not seek to identify the speaker of the quoted text, although the majority of cases in random evaluated samples did represent the patient's speech. Identification of the speaker would clearly prove an interesting avenue for future research.

Despite limitations, this method is a novel approach, as identifying quotations in clinical text has not been a focal point of research to date. We believe that this is a potentially fruitful avenue of investigation, as quotations may have referenced a clinician variable that has not been previously investigated. Furthermore, as indicated, quoted text was relatively common for the subset who had documentation in the comparison periods of interest. Although findings for the primary hypothesis were null in this comparison, the development of an NLP algorithm has provided the means for the creation of a database of quotations across CRIS, generating a much larger sample that might be of interest for future work: both analysing the occurrence or not of quoted text, and potentially the content of such quotations for further characterisation (eg, speech patterns, sentiment).

In conclusion, this study found that there was no difference in the levels of quoted text for individuals at 1–30 days vs 61–90 days prior to a suicide attempt. However, the successful identification of quoted speech within mental healthcare EHRs may have other applications, and there may be fruitful progress to be made in automating the extraction of such text and analysing what the clinician thinks it is important to emphasise from the patient's account.

**Contributors** RD and RS conceived the study design. LJ wrote the paper and analysed the data. LJ also led the development of the quoted speech algorithm with input from AB. RD and RS provided clinical insight on the paper and supervisory guidance. All authors provided critical input for the paper and approved the submission.

**Funding** LJ and RD are part-funded by the NIHR Specialist Biomedical Research Centre for Mental Health at the South London and Maudsley NHS Foundation Trust and Institute of Psychiatry, King's College London. RD is also funded by a Clinician Scientist Fellowship (project e-HOST-IT) from the Health Foundation in partnership with the Academy of Medical Sciences, which also funds AB. RS is part-funded by: (1) the NIHR Specialist Biomedical Research Centre for Mental Health at the South London and Maudsley NHS Foundation Trust and Institute of Psychiatry, King's College London; (2) a Medical Research Council (MRC) Mental Health Data Pathfinder Award to King's College London; and (3) an NIHR Senior Investigator Award.

**Competing interests** RD declares previous research funding received from Janssen. RS has received research support in the last 5 years from Roche, Janssen, GSK and Takeda.

**Patient and public involvement** Patients and/or the public were not involved in the design, or conduct, or reporting, or dissemination plans of this research.

**Patient consent for publication** Not required.

**Ethics approval** Clinical Record Interactive Search, as a data resource for secondary analysis, has Institutional Review Board approval from Oxford C Research Ethics Committee (reference 18/SC/0372).

**Provenance and peer review** Not commissioned; externally peer reviewed.

**Data availability statement** Data are available upon reasonable request. Data sharing statement: data must remain within the SLaM firewall and any requests to access the data can be addressed to cris.administrator@kcl.ac.uk.

**ORCID iD**
Lasantha Jayasinghe http://orcid.org/0000-0003-3907-2645

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
