## [Reviewer comments · BMJ Open]

ARTICLE DETAILS

TITLE (PROVISIONAL)	Clinician- recalled Quoted Speech in Electronic Health Records and Risk of Suicide Attempt: A Case-Crossover Study
AUTHORS	Jayasinghe, Lasantha; Bittar, André; Dutta, Rina; Stewart, Robert

VERSION 1 – REVIEW

REVIEWER	Gregory Simon Kaiser Permanente Washington Health Research Institute Seattle, USA
REVIEW RETURNED	11-Dec-2019

GENERAL COMMENTS	General comment: The general topic is of interest to a broad clinical audience. The hypothesis is novel, but has some support from previous research. While null findings are disappointing, they should certainly be reported. Specific comments 1) I partially disagree with the authors' statements regarding the case-crossover design. That design would be appropriate for identifying within-person associations but not appropriate for identifying between-person associations. Suicide risk almost certainly includes both time-varying (i.e. within-person characteristics) and stable (i.e. between-person) characteristics. The case-crossover method would, by design, overlook the latter. Models predicting suicidal behavior from health records data have typically identified many more long-term or stable risk factors than short-term or time-varying risk factors. This key point deserves clearer explanation and discussion. 2) While the authors' hypothesis concerned providers' quotations of patients' speech, their methods (if I understand them correctly) identified any quoted text from any source. I believe this would have also included quotations from other visit notes. If this is correct, it should be clarified very early in the manuscript and emphasized as a major limitation. Are any data available regarding the proportion of identified quoted text that referred to patients' speech? 3) I was not able to clearly determine what proportion of people (or what proportion of encounters) had any quoted text identified. We might interpret a null finding differently if the hypothesized risk indicator was quite rare or quite common. Table 3 reports quotations per token, but the level of tokenization was not clear (word, phrase, sentence, etc.).
--

REVIEWER	Colin Walsh Vanderbilt University Medical Center USA
REVIEW RETURNED	20-Jan-2020

GENERAL COMMENTS	The authors present a study including regular expression-based means of identifying quoted phrases in electronic health record text with intent to assess its contribution to suicide attempt risk identification. Strengths of the work include reliance on real-world study data, an intent to include an understudied stream of data, and transparent statistics though I have concerns (below) on the overall study design given its impact on inclusion criteria. In its current form, this manuscript is hindered primarily by lack of control for confounding and open questions related to study design.  • Justification for the study design is insufficient and has the important impact of eliminating much of the potential corpus of available text data. 90% of potential data were not used here for lack of a note in the case period and control period. The case-crossover design was described as having the advantage that “individuals serve as their own controls”. This argument needs to be made much more strongly if such a large amount - the vast majority of notes - cannot be used in the remainder of the study. Attempts should also be made to justify why other confounders (clinical workflow, providers, practice and system-level factors) are also accounted for in this design. • The authors are encouraged to speak to the scalability and importance of this method - identifying quotes in clinical text - which is often a preprocessing step in an NLP pipeline but has not been described to my knowledge as the focal point of an NLP approach. • Title is misleading as the corpus is not speech - it is documentation of recollection of speech. The title as written suggests audio or audiovisual recordings might be taking place. The latter domains are an active area of research, so potential for confusion is non-trivial. • Moreover on the point above, documentation of recollected speech might occur within encounters or hours (or days) after encounters. Of time-variant factors that seem important to control for, this one is notable but not mentioned. • The hypothesis requires better justification. A number of factors not mentioned in this manuscript might also contribute to likelihood or frequency of quoted text. For example, I see no control for the provider writing notes, yet clinical documentation by individual providers alone might explain variance of frequency of quoted text. * Univariate analyses are outmoded in this context and do not add to the rigor of the results. A number of studies support this claim including work by Harrell and others, including Nojima et al in this same journal, 2018.
---

VERSION 1 – AUTHOR RESPONSE

Reviewer: 1

General comment: The general topic is of interest to a broad clinical audience. The hypothesis is novel, but has some support from previous research. While null findings are disappointing, they should certainly be reported.

Specific comments

1) I partially disagree with the authors’ statements regarding the case-crossover design. That design would be appropriate for identifying within-person associations but not appropriate for identifying between-person associations. Suicide risk almost certainly includes both time-varying (i.e. within-person characteristics) and stable (i.e. between-person) characteristics. The case-crossover method would, by design, overlook the latter. Models predicting suicidal behavior from health records

data have typically identified many more long-term or stable risk factors than short-term or time-varying risk factors. This key point deserves clearer explanation and discussion.

We accept this comment and have carried out extensive revisions to the Introduction (paragraph 3) which we hope now better frame the rationale for the study and the choice of the case-crossover design. In particular, we present what we hope is a more extensive consideration of between-person and within-person risk factors. We have also reiterated this point in the Discussion (paragraph 4) when describing the generic strengths of the design.

2) While the authors' hypothesis concerned providers' quotations of patients' speech, their methods (if I understand them correctly) identified any quoted text from any source. I believe this would have also included quotations from other visit notes. If this is correct, it should be clarified very early in the manuscript and emphasized as a major limitation. Are any data available regarding the proportion of identified quoted text that referred to patients' speech?

We accept that this is a potential limitation, as the algorithm was indeed not specifically designed to identify the speaker of the quoted speech. However, we have carried out further investigation of the implied speaker in random samples of quoted speech from the control and case periods, finding relatively high proportions that were explicitly quotes from the patient (70.5% and 95.8% respectively). This information has been added to the Methods section (Identification of quoted speech, paragraph 3) has also been added as a limitation to the Discussion (paragraph 6).

3) I was not able to clearly determine what proportion of people (or what proportion of encounters) had any quoted text identified. We might interpret a null finding differently if the hypothesized risk indicator was quite rare or quite common.

We apologise for this oversight and have added this information to the Methods section (in the Study sample description). As the Reviewer will see, the risk indicator was quite common.

Table 3 reports quotations per token, but the level of tokenization was not clear (word, phrase, sentence, etc.).

A footnote has been added to table 3 and table 4 to explain how tokens were derived.

Reviewer: 2

The authors present a study including regular expression-based means of identifying quoted phrases in electronic health record text with intent to assess its contribution to suicide attempt risk identification. Strengths of the work include reliance on real-world study data, an intent to include an understudied stream of data, and transparent statistics though I have concerns (below) on the overall study design given its impact on inclusion criteria. In its current form, this manuscript is hindered primarily by lack of control for confounding and open questions related to study design.

- 1) Justification for the study design is insufficient and has the important impact of eliminating much of the potential corpus of available text data. 90% of potential data were not used here for lack of a note in the case period and control period. The case-crossover design was described as having the advantage that "individuals serve as their own controls". This argument needs to be made much more strongly if such a large amount - the vast majority of notes - cannot be used in the remainder of the study.

As mentioned in the response to Reviewer 1, we have substantially amended the Introduction to consider the rationale for considering within-individual as well as between-individual risk factors. Changes have also been made to the Methods section (Case-crossover design) with an expanded description of the applicability of the case-crossover design and its place in risk factor research. The limitation of the analysed subgroup is highlighted in the Discussion section (paragraph 5).

Attempts should also be made to justify why other confounders (clinical workflow, providers, practice and system-level factors) are also accounted for in this design.

Changes have been made to the Methods section (Time-variant factors) to provide added justification for the design. Text has also been added to the Discussion (strengths paragraph 4) to highlight the further potential advantages suggested.

- 2) The authors are encouraged to speak to the scalability and importance of this method - identifying quotes in clinical text - which is often a preprocessing step in an NLP pipeline but has not been described to my knowledge as the focal point of an NLP approach.

Text has been added to the Discussion (penultimate paragraph) to consider the implications of the NLP development, which we agree has potential much wider applicability.

- 3) Title is misleading as the corpus is not speech - it is documentation of recollection of speech. The title as written suggests audio or audiovisual recordings might be taking place. The latter domains are an active area of research, so potential for confusion is non-trivial.

We agree and have amended the title.

- 4) Moreover on the point above, documentation of recollected speech might occur within encounters or hours (or days) after encounters. Of time-variant factors that seem important to control for, this one is notable but not mentioned.

We acknowledge that this is the case and have made changes to the limitations paragraph in the Discussion.

- 5) The hypothesis requires better justification. A number of factors not mentioned in this manuscript might also contribute to likelihood or frequency of quoted text. For example, I see no control for the provider writing notes, yet clinical documentation by individual providers alone might explain variance of frequency of quoted text.

We accept this point and have expanded on the point in the Discussion (limitations paragraph).

- * 6) Univariate analyses are outmoded in this context and do not add to the rigor of the results. A number of studies support this claim including work by Harrell and others, including Nojima et al in this same journal, 2018.

We note that the paper by Nojima et al (2018) suggests that binary univariate analysis of variables should not be used to select which variables to put into a multivariate regression, but that decisions should be made finally on clinician judgement, rather than p-values. We think there may be a misunderstanding regarding the univariate and multivariate conditional logistic regression results in Table 4. We present all results for univariate variables, and all variables are included in the multivariate regression, regardless of p-value. Likewise, Table 5 presents adjusted associations for all independent variables, which we felt was the most appropriate *a priori* approach. We hope that we have understood the Reviewer's concerns correctly here.

VERSION 2 – REVIEW

REVIEWER	Gregory Simon Kaiser Permanente Washington USA
REVIEW RETURNED	20-Feb-2020

GENERAL COMMENTS	All of my concerns have been addressed.
---